# Truncation and constitutive activation of the androgen receptor by diverse genomic rearrangements in prostate cancer

Christine Henzler[1,*], Yingming Li[2,*], Rendong Yang[1,*], Terri McBride[2,3], Yeung Ho[2], Cynthia Sprenger[4], Gang Liu[4], Ilsa Coleman[5], Bryce Lakely[6], Rui Li[7], Shihong Ma[7], Sean R. Landman[8], Vipin Kumar[8], Tae Hyun Hwang[9], Ganesh V. Raj[7], Celestia S. Higano[5,6,10], Colm Morrissey[6], Peter S. Nelson[5], Stephen R. Plymate[4,6,11] & Scott M. Dehm[2,12]

Molecularly targeted therapies for advanced prostate cancer include castration modalities that suppress ligand-dependent transcriptional activity of the androgen receptor (AR). However, persistent AR signalling undermines therapeutic efficacy and promotes progression to lethal castration-resistant prostate cancer (CRPC), even when patients are treated with potent second-generation AR-targeted therapies abiraterone and enzalutamide. Here we define diverse *AR* genomic structural rearrangements (*AR*-GSRs) as a class of molecular alterations occurring in one third of CRPC-stage tumours. *AR*-GSRs occur in the context of copy-neutral and amplified *AR* and display heterogeneity in breakpoint location, rearrangement class and sub-clonal enrichment in tumours within and between patients. Despite this heterogeneity, one common outcome in tumours with high sub-clonal enrichment of *AR*-GSRs is outlier expression of diverse AR variant species lacking the ligand-binding domain and possessing ligand-independent transcriptional activity. Collectively, these findings reveal *AR*-GSRs as important drivers of persistent AR signalling in CRPC.

[1] Minnesota Supercomputing Institute, University of Minnesota, 117 Pleasant Street Southeast, Minneapolis, Minnesota 55455, USA. [2] Masonic Cancer Center, University of Minnesota, Mayo Mail Code 806, 420 Delaware Street Southeast, Minneapolis, Minnesota 55455, USA. [3] Medical Scientist Training Program, University of Minnesota, Minneapolis, Minnesota 55455, USA. [4] Division of Gerontology and Geriatric Medicine, University of Washington, Seattle, Washington 98104, USA. [5] Fred Hutchinson Cancer Research Center, 1100 Fairview Avenue North, Seattle, Washington 98109, USA. [6] Department of Urology, University of Washington, 1959 Northeast Pacific Street, Box 356510, Seattle, Washington 98195, USA. [7] Department of Urology, University of Texas Southwestern Medical Center, 5323 Harry Hines Boulevard, Dallas, Texas 75390, USA. [8] Department of Computer Science and Engineering, University of Minnesota, 200 Union Street Southeast, Minneapolis, Minnesota 55455, USA. [9] Quantitative Biomedical Research Center, University of Texas Southwestern Medical Center, 2201 Inwood Road, Dallas, Texas 75390, USA. [10] Department of Medicine, University of Washington, 825 Eastlake Avenue East, Seattle, Washington 98109, USA. [11] Geriatric Research Education and Clinical Centers, VA Puget Sound Health Care System, 325 9th Avenue, Box 359625, Seattle, Washington 98104, USA. [12] Departments of Laboratory Medicine and Pathology and Urology, University of Minnesota, Minneapolis, Minnesota 55455, USA. * These authors contributed equally to this work. Correspondence and requests for materials should be addressed to S.M.D. (email: dehm@umn.edu).

Genomic interrogation of castration-resistant prostate cancer (CRPC) tumours has provided a catalogue of mutations and copy number alterations that are associated with disease spread and evolution during androgen receptor (AR)-targeted androgen deprivation therapy (ADT)[1–5]. A consistent finding across these studies is that the androgen/AR axis is the most frequently altered pathway in CRPC. These alterations enable persistent AR transcriptional activity during ADT and thereby drive therapeutic resistance[6–8]. Abiraterone and enzalutamide are second-generation agents that can re-target the AR pathway after failure of ADT, by more effectively inhibiting intratumoral steroidogenesis or inhibiting androgens from binding to the AR ligand-binding domain. However, as with standard ADT, resistance to these agents invariably occurs and AR continues to drive tumour progression. Thus, abiraterone- and enzalutamide-resistant CRPC has emerged as a contemporary clinical challenge[9,10].

The most common documented alterations in the AR pathway in CRPC, occurring in ~60% of tumours, are amplification or mutation of the AR gene. There are efforts underway to develop approaches for monitoring AR amplification or mutation, which could inform more precise use of first- and second-generation AR-targeted therapies, indicate the need for alternative therapies such as taxanes, or assist in prioritization of patients for clinical trials. For example, detection of AR amplification or mutations in cell-free DNA isolated from blood of CRPC patients is associated with resistance to abiraterone and enzalutamide[11]. Moreover, detection of these alterations at baseline appears to predict primary resistance[12].

Expression of AR messenger RNA (mRNA) splicing variants (AR-Vs) lacking the AR ligand-binding domain has emerged as an additional mechanism of resistance to first- and second-generation AR-targeted therapies. In particular, detection of AR-V7 mRNA in circulating tumour cells from CRPC patients treated with abiraterone or enzalutamide is associated with primary resistance and reduced overall survival[13,14]. Additional AR-Vs have also been reported in CRPC models, clinical tissues and circulating tumour cells[15–22]. Functionally, some AR-Vs have been shown to promote resistance by engaging AR chromatin-binding sites and driving the AR transcriptional programme in a constitutive, ligand-independent manner[19,23]. However, the importance of AR-Vs as a clinically relevant resistance mechanism has been controversial, because large-scale studies of CRPC have shown that mRNA levels of AR-V7 and other known AR-Vs are low relative to full-length AR[3]. Moreover, AR-V7:AR mRNA expression ratios observed in CRPC do not appear to be increased relative to therapy-naive prostate cancer, normal prostate tissue or even non-prostate tissues[3,24,25].

In CRPC models where AR-Vs have been shown to promote resistance, expression levels of AR-Vs relative to full-length AR are high and have been linked to specific AR gene rearrangement events[26–28]. Although these AR gene rearrangements are well-characterized in these models, the relevance of AR gene rearrangements to clinical CRPC has been unclear. To address this, in this study we conduct a deep sequencing analysis of AR in tissues derived from metastatic CRPC, localized CRPC and therapy-naive prostate cancer. Our results demonstrate that AR gene rearrangements are frequent and diverse in clinical disease. By integrating these findings with AR expression data, we demonstrate that AR gene rearrangements with high allelic enrichment drive outlier expression of unique AR-V species with constitutive transcriptional activity and protein structures resembling AR-V7. In conclusion, AR gene rearrangements represent an important mechanism of AR-V expression in clinical CRPC.

## Results

### AR-GSRs in clinical CRPC.
The 183 kb AR gene locus is located on the X chromosome and contains multiple repetitive DNA stretches including long- and short-interspersed nuclear elements. We developed a liquid-phase AR bait panel with enhanced coverage features that we hypothesized would provide greater sensitivity for detection of structural aberrations in this high repeat-content locus (Supplementary Fig. 1 and Supplementary Table 1). Using this enhanced AR bait panel, we performed targeted paired-end ($2 \times 150$ bp) Illumina sequencing of DNA (AR DNA-seq) from 30 soft tissue metastases (Supplementary Data 1). These tumours were obtained by rapid autopsy of 15 CRPC patients with diverse clinical and treatment histories, and 2 anatomically distinct tumours were studied per patient (Supplementary Tables 2 and 3). Average per-base sequence coverage of the AR gene ranged from 283X to 1293X, with 83–89% of AR covered (Supplementary Table 4). This represented an improvement over previous targeted AR DNA-seq studies where AR coverage was 75% (refs 27,28). Analysis of AR DNA-seq data demonstrated that 12/30 tumours (6/15 patients) in this metastatic CRPC cohort displayed AR gene amplification, which was defined as normalized coverage of the AR gene >1 when compared with normalized coverage at a set of control genomic regions (Fig. 1a and Supplementary Table 4). This frequency of AR gene amplification is similar to previous DNA-seq and array-based studies of CRPC metastases[2–4]. In addition, 6/30 CRPC metastases (3/15 patients) displayed missense mutations in the AR gene with variant allele frequencies ranging from 38 to 85% (T878A in both metastases from subject C-1, T878A in both metastases from patient C-10 and L702H in both metastases from patient C-8; Fig. 1a and Supplementary Table 5). These are hotspot mutations occurring in the AR ligand-binding domain in CRPC and are known to drive inappropriate agonist responses of the AR to various ligands, including antiandrogens[1,3].

Next, we analysed AR DNA-seq data with a structural variant detection pipeline to develop a list of candidate AR genomic structural rearrangements (AR-GSRs), defined as rearrangement events having at least one breakpoint mapping within the AR gene body (Supplementary Data 2). This approach indicated that AR-GSRs were frequent in metastatic CRPC, with 10/30 metastases (6/15 patients) displaying at least one AR-GSR event (Fig. 1a). AR-GSRs occurred in 3/18 CRPC metastases that were AR copy neutral and at higher frequency (7/12) in CRPC metastases that displayed AR amplification ($P = 0.0450$, two-tailed Fisher's exact test). Although AR-GSRs did not occur in the six CRPC metastases with AR missense mutations, tests for mutual exclusivity between AR-GSRs and AR missense mutations did not reach statistical significance (0/6 versus 10/24, $P = 0.0741$, two-tailed Fisher's exact test). Notably, AR-GSR-positive metastases C-6A, C-6B and C-9A did not harbour AR amplification or mutation events. This showed that AR-GSR positivity could re-classify a subset of CRPC metastases that would otherwise have been deemed to harbour a normal AR gene.

To understand the context of this new class of AR gene alteration, we performed AR DNA-seq with six additional CRPC specimens obtained from transurethral resection of the prostate (TURP) procedures and 21 hormone-naive prostate cancer specimens obtained from prostatectomies (Supplementary Table 6). Similar to the frequency observed in CRPC metastases, AR-GSRs were detected in two of six localized CRPC TURP specimens (Fig. 1b). Conversely, all hormone-naive prostate cancer specimens were AR-GSR negative (Fig. 1b). Collectively, these data indicated that AR-GSRs were restricted to CRPC ($P = 0.0021$, two-tailed Fisher's exact test).

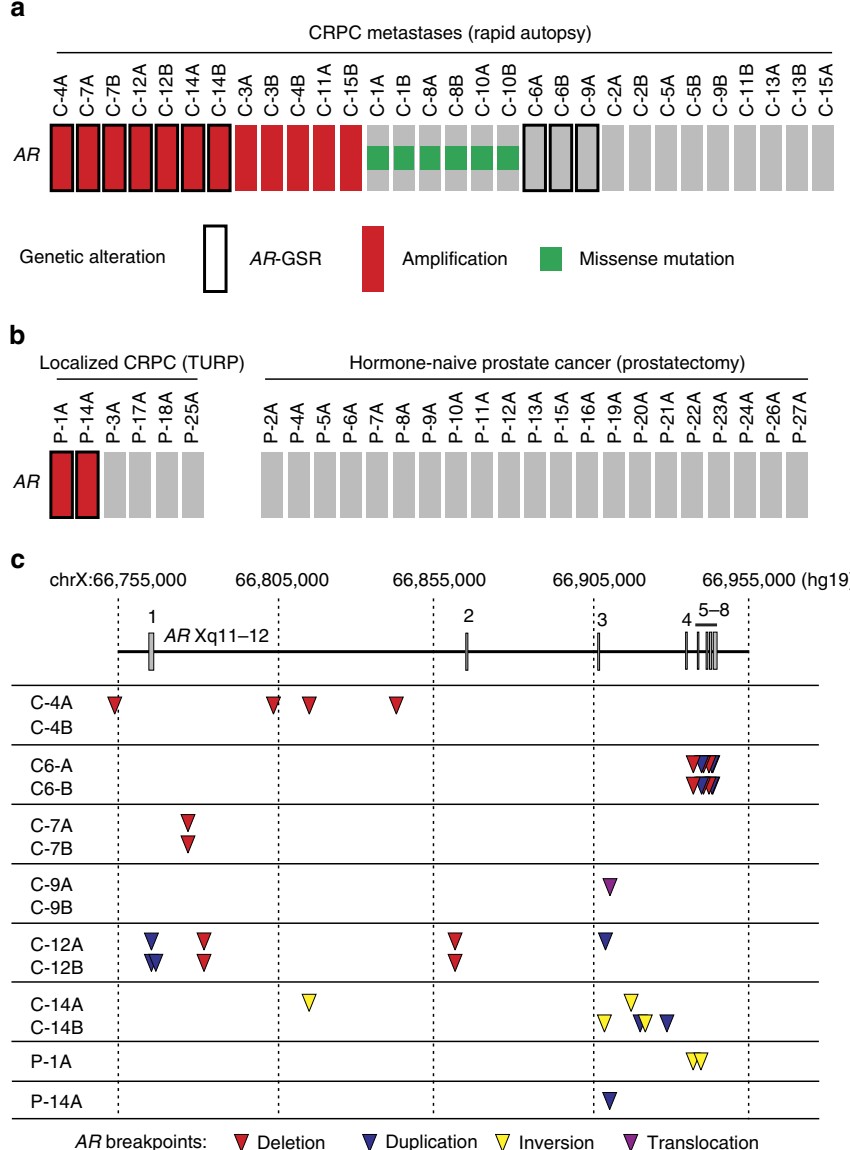

**Figure 1 | Frequent and diverse *AR* gene structural rearrangements in CRPC metastases.** (**a**) Oncoprint illustrating the presence of *AR* amplification, missense mutations or *AR*-GSRs in 30 metastases from 15 rapid autopsy subjects. (**b**) Oncoprint illustrating the presence of *AR* amplification or *AR*-GSRs in 6 localized CRPC specimens obtained by TURP and 21 hormone-naive prostate cancer specimens. (**c**) Map of *AR*-GSR breakpoint locations within the *AR* gene body for nine *AR*-GSR patients. Coloured triangles represent break fusion junction locations, with the colour specifying whether that break fusion junction was due to a discrete deletion (red), duplication (blue), inversion (yellow) or translocation (purple) event. Genome coordinates are genome build GRCh37/hg19. Locations of *AR* exons 1–8 are shown.

**AR-GSRs arise from diverse rearrangement events**. The precise locations of *AR*-GSR break fusion junctions were variable across the *AR* gene body and associated with diverse deletion, duplication, inversion and translocation events (Fig. 1c). We used targeted PCR to isolate the break fusion junctions for each of the 21 candidate *AR*-GSRs detected in CRPC metastases and confirmed the sequence of each PCR product by Sanger sequencing (Supplementary Figs 2–7). Inspection of the discrete *AR*-GSRs revealed that every break fusion junction contained a signature of non-homologous end joining and break fusion junctions were frequently located within long- and short-interspersed nuclear elements (Supplementary Figs 2–7). Of the 12 CRPC tumours that were positive for an *AR*-GSR event, 7 harboured multiple *AR*-GSRs (Fig. 1c). Remarkably, no two patients displayed the same *AR*-GSR breakpoint location or break fusion junction signature (Fig. 1c), even when compared with

*AR*-GSRs reported previously in CRPC models[26–28]. However, independent metastases analysed from rapid autopsy subjects C-6, C-7 and C-12 shared identical *AR*-GSR break fusion junction signatures, suggesting intra-patient clonality (Fig. 1c and Supplementary Figs 3, 4 and 6).

**AR gene copy number and AR-V7 mRNA expression levels**. To understand the functional impact of these *AR*-GSRs, we first tested whether their presence in CRPC metastases was associated with expression of AR-V7. This was based on previous work showing that high levels of AR-V7 expression relative to full-length AR was associated with *AR*-GSRs in the 22Rv1 and CWR-R1 CRPC cell lines[26,27]. Expression of full-length AR and AR-V7 mRNA species was detectable by reverse transcriptase PCR (RT–PCR) in all CRPC metastases used for AR DNA-seq

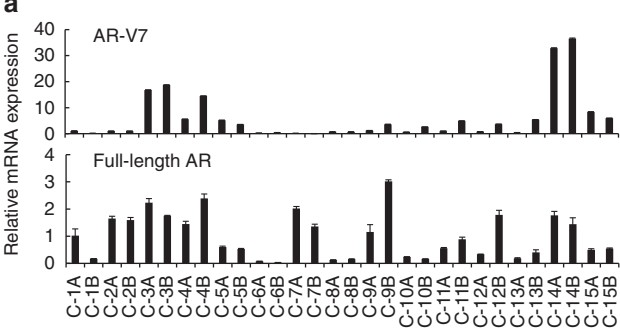

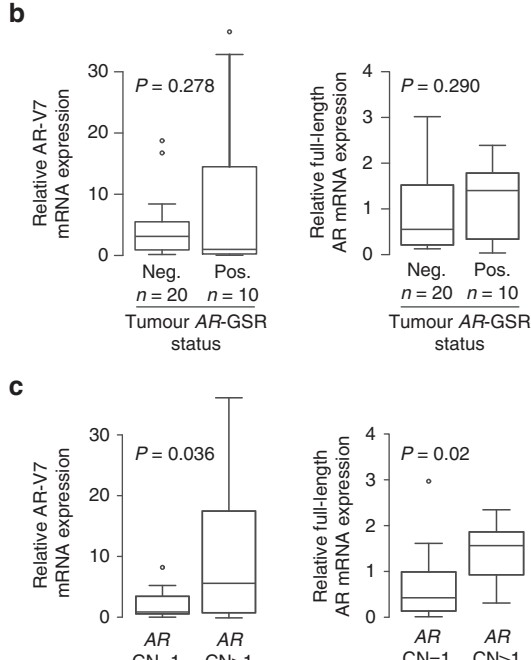

**Figure 2 | Association of AR-V7 mRNA expression levels with AR gene copy number.** (**a**) mRNA expression of AR-V7 (top) and full-length AR (bottom) was assessed by quantitative RT–PCR. Levels are shown relative to a housekeeping gene (*RPL13A*), with expression in tumour C-1A set to 1. Data represent mean ± s.d. from three repeated experiments each consisting of independent RT and PCR reactions from the same original RNA sample ($n = 3$). Relative mRNA expression levels of AR-V7 (left) or full-length AR (right) were compared in tumours that (**b**) were positive (Pos.) or negative (Neg.) for an *AR* gene structural rearrangement (*AR*-GSR) event or (**c**) had *AR* gene copy number (CN) ≥1. Centre lines show the medians; box limits indicate the 25th and 75th percentiles; whiskers extend 1.5 times the interquartile range from the 25th and 75th percentiles and outliers are represented by dots. *P*-values were determined by one-tailed Mann–Whitney *U*-test.

analysis, but AR-V7 displayed a broader range (Fig. 2a). However, there did not appear to be differences in the relative levels of AR-V7 or full-length AR mRNA in tumours that were positive for an *AR*-GSR event compared with tumours that were negative for an *AR*-GSR event (Fig. 2b). Further, no mutations were detected in splice donor or acceptor sites of canonical *AR* exons or cryptic exon (CE) 3, the 3′-terminal exon spliced in AR-V7 mRNA. Conversely, levels of AR-V7 and full-length AR were significantly higher in tumours that displayed an increase in *AR* gene copy number compared with tumours that displayed a single *AR* gene copy (Fig. 2c). These data indicated that *AR* gene dosage,

as opposed to the presence of *AR*-GSRs, was the main determinant of AR-V7 mRNA levels in CRPC metastases. However, it should be noted that tumours C-14A and C-14B had the highest levels of AR-V7 mRNA across this cohort of CRPC metastases and also displayed inversion and duplication events involving the intron 3 region where the AR-V7 3′-terminal exon CE3 is located (Supplementary Fig. 7).

Next, we considered previous work demonstrating that *AR*-GSR-positive cells in prostate cancer cell lines and patient-derived xenografts are often sub-clonal, but represent the tumour cell fractions expressing high levels of AR-Vs and displaying androgen-independent growth[19,27,28]. Under this scenario, sub-clonality of *AR*-GSRs would confound efforts to link discrete *AR*-GSR events to gene expression data from bulk tumour samples, in particular for a broadly expressed mRNA such as AR-V7. To address this issue, we used the SHEAR algorithm[29] to estimate variant allele fraction of *AR*-GSRs in heterogenous samples (Table 1). This approach revealed a broad range of sub-clonal heterogeneity for the detected *AR*-GSRs, indicating that cells harbouring *AR*-GSRs existed as minor populations in some of the tumour samples, but major populations in others. Variant allele fractions for enriched *AR*-GSRs were similar to variant allele fractions for *AR* missense mutations (Table 1 and Supplementary Table 5). Examination of sections of CRPC metastases adjacent to the sections used for genomic DNA isolation revealed high cancer cell purity, indicating this sub-clonality was mainly due to variable percentages of tumour cells harbouring *AR*-GSRs or missense mutations (Supplementary Table 2). Consequently, we concluded that a candidate-by-candidate approach would be required to understand the potential impact of these *AR*-GSRs on AR expression. We therefore prioritized rapid autopsy subjects C-6, C-9 and C-12, as they all harboured tumours with *AR*-GSRs at variant allele fractions >10% (Table 1).

**AR-GSR in patient C-6 associated with ARv567es.** Quantitative RT–PCR specific for mRNA encoding ARv567es, a constitutively active AR-V species that has been linked to underlying *AR*-GSRs in CRPC xenograft models[21,28], revealed a dramatic outlier pattern of expression in tumours C-6A and C-6B (Fig. 3a). Notably, three discrete *AR*-GSRs in rapid autopsy subject C-6 were present at high variant allele frequency (Table 1), concentrated in the region of the *AR* gene encoding the AR ligand-binding domain (Fig. 1c), and clonal between tumours C-6A and C-6B (Supplementary Fig. 3). Interestingly, tumours C-6A and C-6B also displayed the lowest levels of full-length AR mRNA expression across the cohort of CRPC metastases (Fig. 2a). This suggested the *AR*-GSRs detected in this patient were driving a shift in AR splicing towards ARv567es. We therefore interrogated three tumour sites from this patient (Fig. 3b) by immunohistochemistry with an antibody specific for the unique COOH-terminal extension in ARv567es protein (Fig. 3c and Supplementary Fig. 8). This approach revealed uniformly intense nuclear expression of ARv567es protein at all sites studied, including a pelvic bone metastasis (Fig. 3d).

Based on these findings, we reflected on the possible mechanism by which three discrete *AR*-GSRs that appeared to share overlap with one another could co-exist in tumours harbouring a single *AR* gene copy (Supplementary Fig. 3a,b). One explanation could be that each *AR*-GSR was restricted to a separate clone in the tumour. In this scenario, differential AR gene copy number profiles would be expected for the two distinct tumour sites as a result of differential enrichment of these sub-clones. However, AR DNA-seq read coverage in tumours C-6A and C-6B was indistinguishable (Supplementary Fig. 3c).

**Table 1 | Summary of variant allele fractions of *AR*-GSRs in prostate cancer.**

| Patient | Tumour | Tumour type | *AR*-GSR | Break fusion junction coordinates (hg19) | Variant allele fraction (SHEAR) |
|---------|--------|-------------|----------|-------------------------------------------|--------------------------------|
| C-4 | A | CRPC met | Deletion 1 | chrX:66,804,042 (+)/chrX:66,816,507 (+) | 7.32% |
| | A | CRPC met | Deletion 2 | chrX:66,754,961 (+)/chrX:66,843,254 (+) | 7.88% |
| C-6 | A | CRPC met | Deletion 1 | chrX:66,934,778 (+)/chrX:66,942,396 (+) | 46.99% |
| | A | CRPC met | Duplication | chrX:66,942,924 (+)/chrX:66,939,551 (+) | 74.15% |
| | A | CRPC met | Deletion 2 | chrX:66,940,040 (+)/chrX:66,943,474 (+) | 42.68% |
| | B | CRPC met | Deletion 1 | chrX:66,934,778 (+)/chrX:66,942,396 (+) | 43.45% |
| | B | CRPC met | Duplication | chrX:66,942,924 (+)/chrX:66,939,551 (+) | 49.19% |
| | B | CRPC met | Deletion 2 | chrX:66,940,040 (+)/chrX:66,943,474 (+) | 38.23% |
| C-7 | A | CRPC met | Deletion | chrX:49,150,734 (+)/chrX:66,779,010 (+) | 4.26% |
| | B | CRPC met | Deletion | chrX:49,150,734 (+)/chrX:66,779,010 (+) | 5.40% |
| C-9 | A | CRPC met | Translocation | chrX:66,909,163 (+)/TTTAG/chr11:79,397,735 (+) | 13.74% |
| C-12 | A | CRPC met | Deletion | chrX:66,786,453 (+)/chrX:66,862,260 (+) | 13.13% |
| | A | CRPC met | Duplication 1 | chrX:66,765,788 (+)/chrX:66,738,768 (+) | 4.71% |
| | A | CRPC met | Duplication 2 | chrX:66,909,930 (+)/chrX:66,530,990 (+) | 33.25% |
| | B | CRPC met | Deletion | chrX:66,786,453 (+)/chrX:66,862,260 (+) | 46.28% |
| | B | CRPC met | Duplication 1 | chrX:66,765,788 (+)/chrX:66,738,768 (+) | 25.87% |
| | B | CRPC met | Duplication 3 | chrX:66,767,448 (+)/chrX:66,026,083 (+) | 5.66% |
| C-14 | A | CRPC met | Inversion 1 | chrX:65,828,372 (−)/chrX:66,816,576 (+) | 4.52% |
| | A | CRPC met | Inversion 2 | chrX:66,919,966 (+)/chrX:67,012,163 (−) | 1.62% |
| | B | CRPC met | Inversion 3 | chrX:66,908,944 (−)/66,922,090 (+) | 1.86% |
| | B | CRPC met | Duplication | chrX:66,929,712 (+)/chrX:66,921,594 | 2.65% |
| P-1 | TURP | local CRPC | Inversion | chrX:66,935,187 (+)/chrX:66,938,063 (−) | not determined/imprecise |
| P-14 | TURP | local CRPC | Duplication | chrX:67,337,657 (+)/chrX:66,909,110 (+) | 2.19% |

*AR*-GSR, *AR* genomic structural rearrangement; CRPC, castration-resistant prostate cancer; met, metastases; PC, prostate cancer; TURP, transurethral resection of the prostate.

Next, we considered the alternative hypothesis that these three *AR*-GSRs co-existed within the same cell. This scenario would require the *AR*-GSRs to co-exist on the same *AR* allele, as these tumours harboured a single *AR* gene copy (Fig. 1a and Supplementary Table 4). We envisioned a model wherein *AR* deletions 1 and 2 could have occurred in the context of an *AR* allele that had undergone a prior tandem duplication event, which would disrupt the architecture of the entire *AR* exon 5–7 segment encoding the AR ligand-binding domain (Fig. 3e). To test this hypothesis, we performed PCR with tumour DNA from three distinct tumours in rapid autopsy subject C-6, using primers that flanked all three *AR*-GSR break fusion junctions. This strategy yielded a 1.3 kb PCR product at all tumour sites (Fig. 3f). Sanger sequencing revealed that each of these PCR products harboured all three *AR*-GSR break fusion junction signatures that had been detected by AR DNA-seq analysis. Collectively, these findings demonstrated that CRPC in rapid autopsy subject C-6 consisted of a dominant clone harbouring a single *AR* allele with a complex *AR*-GSR driving synthesis of the constitutively active ARv567es protein.

**AR-GSR in tumour C-12A underlies tumour-specific AR-Vs**. Investigations of tissue from rapid autopsy patient C-6 indicated that certain *AR*-GSRs may be associated with outlier, subject-specific patterns of AR-V mRNA expression (that is, a specific AR-V species may be highly expressed in a single subject, but not other subjects). To test this scenario further, we next focused on subject C-12, where two tumour sites harboured a series of shared and exclusive *AR* deletion and duplication events (Fig. 1b and Supplementary Fig. 6), several of which were highly enriched in the tumour sub-clonal architecture (Table 1). Co-occurrence of these *AR*-GSRs with *AR* gene amplification pointed to a complexity of *AR* gene structures in this subject (Fig. 1a), who had been treated with and developed resistance to abiraterone (Supplementary Table 3). Despite *AR* gene amplification, levels of full-length AR and AR-V7 mRNA in tumour C-12A were low

relative to tumour C-12B and to all other tumours in this cohort (Fig. 2a). Interestingly, tumour C-12A harboured a unique tandem duplication with a break fusion junction mapping between *AR* exon 3 and exon CE3 (the 3′-terminal exon in AR-V7; Fig. 4a). We hypothesized this *AR*-GSR event had the potential to cause *AR* splicing alterations downstream of *AR* exon 3.

To test this hypothesis, we performed Illumina paired-end RNA sequencing (RNA-seq) of tumours C-12A and C-12B, and mapped RNA-seq reads against a C-12A-specific genome assembly harbouring this 379 kb *AR*-GSR duplication event (Fig. 4a). We created bins to represent *AR* exons and introns that would exist within this *AR* architecture, and generated heatmaps to quantify and visualize *AR* mRNA splicing between these bins (Fig. 4b). The diagonals of these heatmaps revealed strong signals for canonical *AR* mRNA splicing events in both tumours. However, the heatmap reflecting C-12A splicing displayed an abundance of splice junctions occurring between *AR* exon 3 and sequences upstream of the *AR* gene locus, as well as splice junctions occurring within the *AR* upstream region (Fig. 4b). Splicing between *AR* exon 3 and upstream exons in tumour C-12A was also detected using the Comrad algorithm[30] designed for identifying fusion transcripts expressed from underlying genomic rearrangements (Supplementary Data 3).

To understand these unusual AR splicing events in more detail, we inspected RNA-seq coverage and splicing junctions. Strikingly, expression of exons was apparent in tumour C-12A but not C-12B due to mRNA splicing events between *AR* exon 3 and these *AR* upstream exons (named exons '4-ups', '5a-ups', '5b-ups' and '5c-ups' to denote their relative positions within spliced AR-V mRNAs but their normal genomic locations upstream of *AR*; Fig. 4c,d and Supplementary Fig. 9). Assembly of the expressed mRNAs and translation of their protein products revealed a set of three AR-Vs nearly identical to previously characterized AR-V species, containing the transcriptionally active AR NH$_2$-terminal domain (NTD) and AR DNA-binding domain (DBD) core (Fig. 4e). These AR-Vs had COOH-terminal

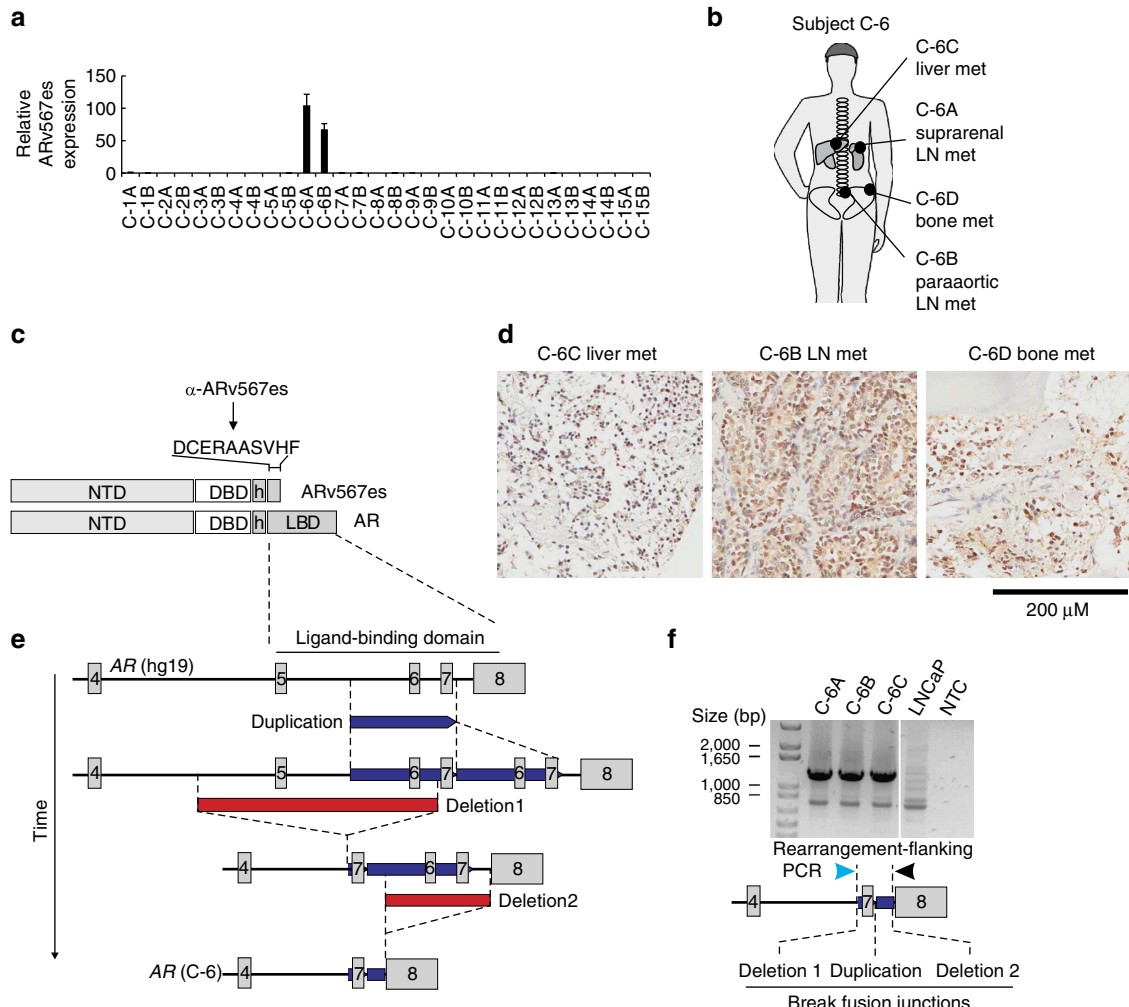

**Figure 3 | Outlier expression of ARv567es in subject C-6 harbouring a tumour clone with a complex *AR*-GSR.** (**a**) mRNA expression of ARv567es was assessed by quantitative RT–PCR. Levels are shown relative to a housekeeping gene (*RPL13A*), with expression in tumour C-1A set to 1. Data represent mean ± s.d. from three repeated experiments, each consisting of independent RT and PCR reactions from the same original RNA sample (*n* = 3). (**b**) Location of metastatic sites analysed for subject C-6. (**c**) Location and sequence of the unique epitope recognized by a rabbit monoclonal antibody specific for ARv567es. (**d**) Three tumour sites from subject C-6 were stained by immunohistochemistry with an antibody specific for ARv567es, revealing strong nuclear expression. (**e**) Model for occurrence of a multi-step rearrangement affecting the sole *AR* gene copy in subject C-6. (**f**) PCR validating existence of the multi-step *AR* rearrangement illustrated in **e** in genomic DNA from three tumour sites from subject C-6. Sanger sequencing of PCR products revealed that all three break fusion junction signatures (duplication and deletions 1 and 2) existed on the same DNA molecule.

extensions of variable length and sequence, with a notable feature being VRRGR motifs adjacent to the AR DBD. Positively charged amino acids in the VRRGR motif aligned with ARKLK and EKFRM motifs from AR and AR-V7, respectively, which constitute the second half of a strong bipartite nuclear localization signal[31]. As would be predicted from this set of features, these AR-Vs had molecular weights similar to AR-V7 and could drive transcriptional activation of AR-responsive promoter reporter constructs in an androgen-independent and enzalutamide-resistant manner in multiple cell lines (Fig. 4f,g and Supplementary Fig. 10). In addition, infection of LNCaP cells with lentiviral vectors harbouring these AR-Vs resulted in androgen-independent proliferation at lower virus titres, but suppression of proliferation at higher virus titres (Fig. 4h and Supplementary Fig. 13a). These results closely resembled the known biphasic effects of AR-V7 and ARv567es expression on LNCaP cell proliferation due to biphasic regulation of proliferation-associated genes[19,31]. From these data, we concluded that an *AR*-GSR enriched in the sub-clonal architecture of tumour C-12A was

responsible for expression of a tumour-specific set of AR-V species that could support constitutive, ligand-independent AR transcriptional activity and prostate cancer cell growth.

**A 3′-terminal *AR* exon from chromosome 11 in tumour C-9A.** Tumour C-9A displayed a translocation that fused a region of chromosome 11 downstream of *AR* exon 3 (Fig. 5a and Supplementary Fig. 5). We hypothesized that *AR* mRNA splicing alterations specific to this *AR*-GSR may be detectable using our strategy of mapping bulk tumour RNA-seq reads to a rearranged genome assembly (Fig. 5a). Indeed, inspection of heatmaps, RNA-seq coverage and splice junction data revealed that tumour C-9A, but not tumour C-9B, expressed an *AR* mRNA species with a splice junction between exon 3 and a 3′-terminal exon recruited from chromosome 11 (Fig. 5b–d and Supplementary Fig. 11). This *AR* exon 3/chr11 splice junction was also detected using the Comrad algorithm (Supplementary Data 3). Assembly of the resulting *AR* mRNA species and translation of the protein

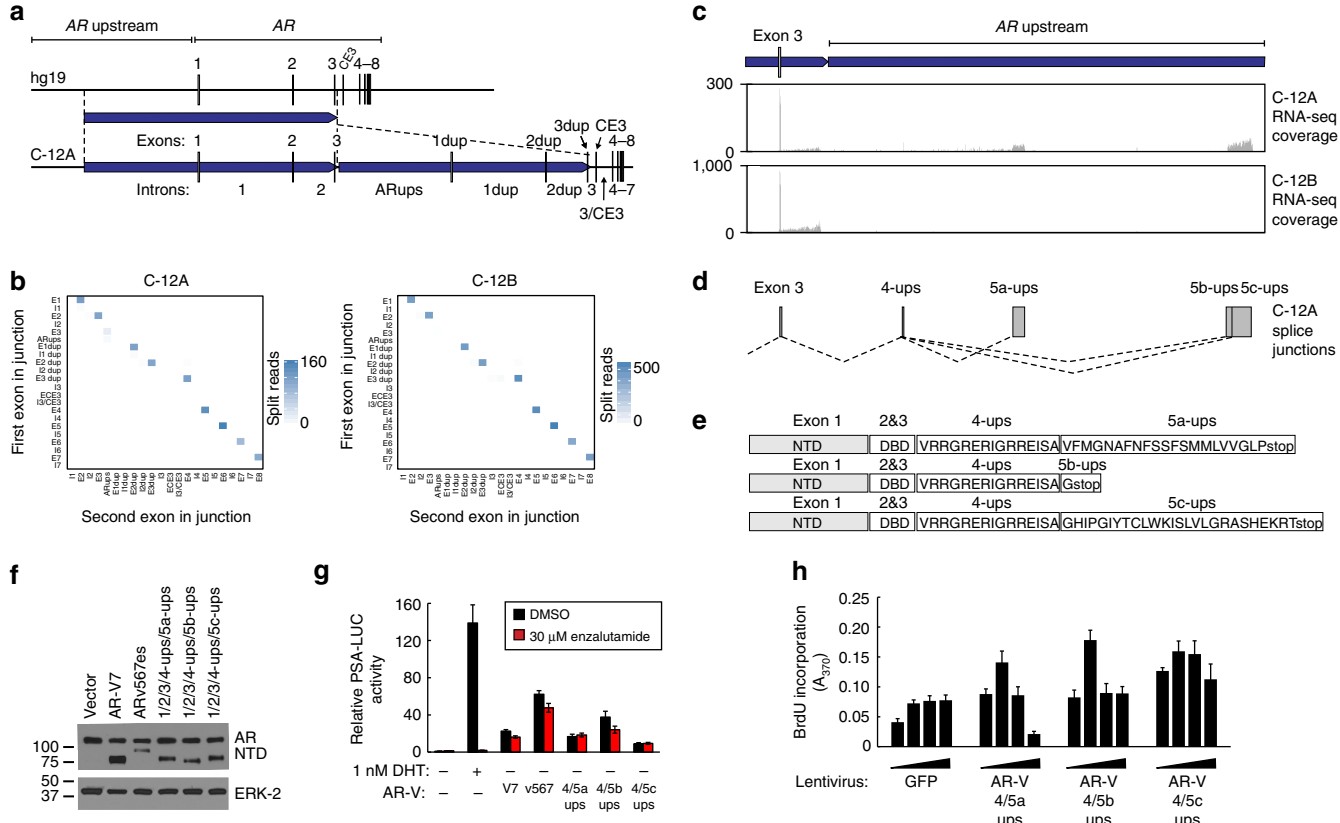

**Figure 4 | An *AR*-GSR in tumour C-12A drives expression of a set of duplication-dependent AR-V mRNAs. (a)** Schematic of *AR* exon and intron organization resulting from the 379 kb tandem duplication in tumour C-12A. **(b)** Heatmaps representing numbers of RNA-seq reads from tumours C-12A and C-12B spanning exon/exon, exon/intron and intron/intron boundaries of the *AR* architecture defined in **a**. Each exon and intron was considered as a discrete 'bin', with each pixel representing counts of RNA-seq splice junctions starting in the first bin (first exon in junction) and ending in the second bin (second exon in junction). **(c)** RNA-seq read coverage from tumour C-12A within the *AR* upstream region that becomes situated downstream of *AR* exon 3 as a result of the 379 kb tandem duplication. RNA-seq read coverage for tumour C-12B, which is negative for this tandem duplication is shown as a control. **(d)** Novel AR splice junctions occurring in tumour C-12A. Exons are named '4-ups', '5a-ups', '5b-ups' and '5c-ups' to denote their splicing order (4 or 5) in the AR-V mRNA and their genomic locations upstream from AR in the reference genome architecture. **(e)** Translation of three novel AR-V mRNA species uniquely expressed in tumour C-12A consisting of the AR NTD, DBD and unique COOH-termini. **(f,g)** LNCaP cells were transfected with a PSA promoter/enhancer-luciferase reporter and expression vectors encoding AR-V7, ARv567es or AR-Vs from tumour C-12A. Cells were treated with 1 nM dihydrotestosterone (DHT), 30 μM enzalutamide (enz) or DMSO (vehicle control) as indicated and subjected to **(f)** western blotting with antibodies specific for the AR NTD or ERK-2 (loading control), or **(g)** Luciferase assay. Data represent mean ± s.e.m. from three biological replicate experiments, each performed in duplicate (*n* = 6). **(h)** LNCaP cells were infected with a range of titres of lentivirus encoding GFP (control) or AR-Vs from tumour C-12A and assayed for proliferation by BrdU incorporation assay. Data represent mean ± s.e.m. from two biological replicate experiments, each performed in triplicate (*n* = 6).

product revealed an AR-V species consisting of the AR NTD/DBD core and a 13 amino acid COOH-terminal extension with a VGKTK motif resembling the AR nuclear localization signal (Fig. 5e). This AR-V had a molecular weight indistinguishable from AR-V7 and displayed transcriptional activity that was androgen-independent and enzalutamide-resistant (Fig. 5f,g and Supplementary Fig. 12). In addition, expression of this AR-V could promote androgen-independent growth of LNCaP cells in a biphasic manner (Fig. 5h and Supplementary Fig. 13b) Thus, similar to patients C-6 and C-12, an *AR*-GSR enriched in the sub-clonal architecture of tumour C-9A was responsible for expression of a constitutively active AR-V species that could promote androgen-independent growth of prostate cancer cells.

**Not all detected *AR*-GSRs underlie tumor-specific AR-Vs.** Our integrative analysis indicated that tumours harbouring *AR*-GSRs at high variant allele fractions expressed tumour-specific AR-V

mRNAs. Conversely, reconstruction and validation of certain *AR*-GSR events present at low variant allele fractions revealed that some of the resultant *AR* gene architectures may not be compatible with AR-V, or even full-length AR, expression. For example, tumours C-4A, C-7A and C-7B all harboured deletions encompassing *AR* exon 1, which encodes the transcriptionally active AR NTD. It is noteworthy that these potentially deleterious events occurred in tumours harbouring multiple *AR* alleles due to *AR* amplification (Fig. 1a). To test the correspondence between *AR*-GSRs and AR-Vs further, we mapped bulk tumour RNA-seq reads from tumours C-14A and C-14B to genome assemblies reflecting the inversions and duplications observed in these tumours (Supplementary Fig. 7). We also performed integrative analysis of RNA-seq and *AR*-GSR data using Comrad[30]. We did not observe evidence for any AR splicing alterations arising due to these *AR*-GSRs (Supplementary Data 3). Collectively, these results indicate that only a subset of *AR*-GSR events are associated with tumour-specific AR-V expression.

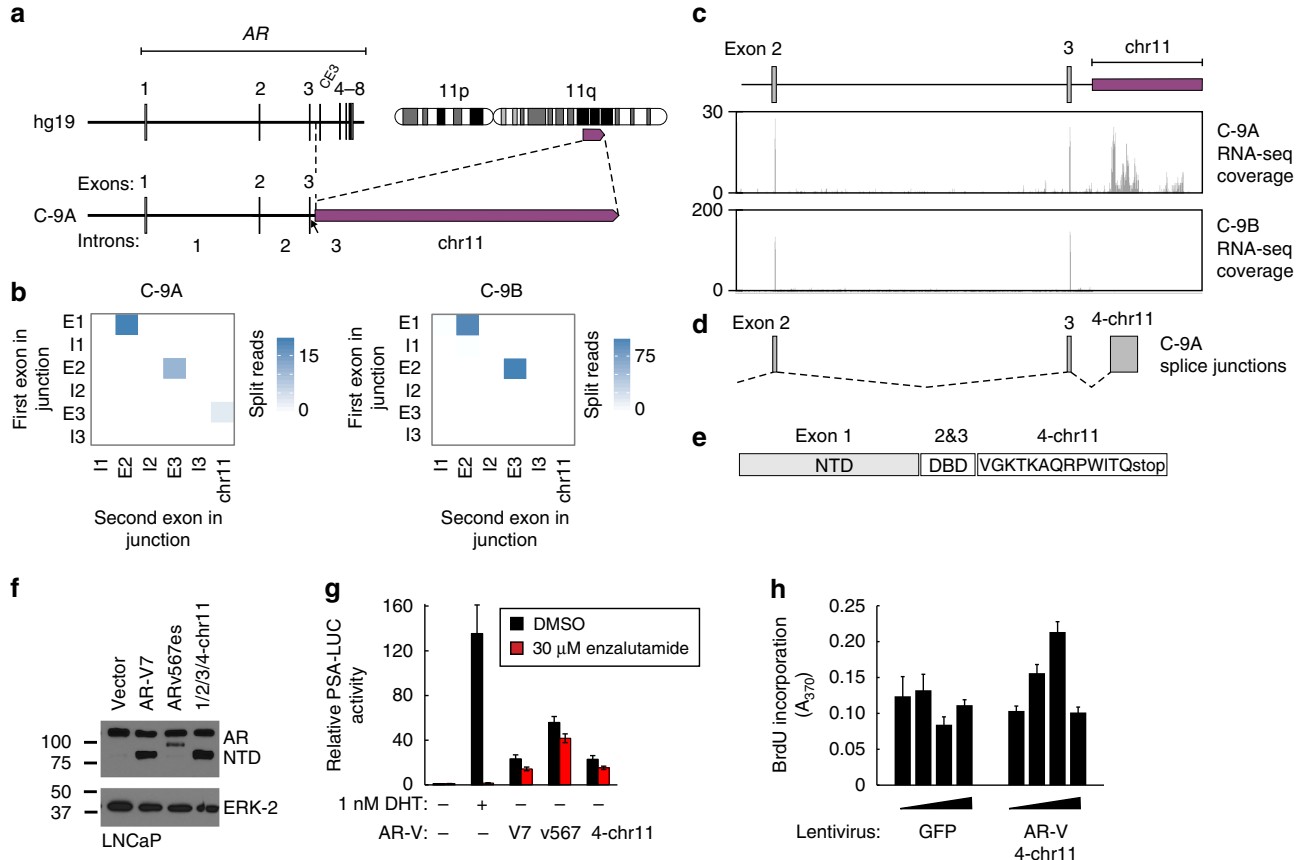

**Figure 5 | An *AR*-GSR in tumour C-9A drives expression of a translocation-dependent AR-V mRNA. (a)** Schematic of *AR* exon and intron organization resulting from the AR:chr11 translocation in tumour C-9A. **(b)** Heatmaps representing numbers of RNA-seq reads from tumours C-9A and C-9B spanning exon/exon, exon/intron and intron/intron boundaries of the AR architecture defined in **a**. Each exon and intron was considered as a discrete 'bin', with each pixel representing counts of RNA-seq splice junctions starting in the first bin (first exon in junction) and ending in the second bin (second exon in junction). **(c)** RNA-seq read coverage from tumour C-9A within the chromosome 11 region that becomes situated downstream of AR exon 3 as a result of the translocation. RNA-seq read coverage from tumour C-9B, which is negative for this rearrangement, is shown as a control. **(d)** Novel AR splice junctions occurring in tumour C-9A. **(e)** Translation of a novel AR-V mRNA species uniquely expressed in tumour C-9A consisting of the AR NTD, DBD and a unique COOH terminus. **(f,g)** LNCaP cells were transfected with a PSA promoter/enhancer-luciferase reporter and expression vectors encoding AR-Vs as indicated. The AR-V encoded by splicing of a novel AR exons from chromosome 11 is indicated in bold. Cells were treated with dihydrotestosterone (DHT), enzalutamide (enz) or DMSO (vehicle control) as indicated and subjected to **(f)** western blotting with antibodies specific for the AR NTD or ERK-2 (loading control), or **(g)** Luciferase assay. Data represent mean ± s.e.m. from three biological replicate experiments, each performed in technical duplicate (*n* = 6). **(h)** LNCaP cells were infected with a range of titres of lentivirus encoding GFP (control) or the unique AR-V expressed in tumour C-9A and assayed for proliferation by BrdU incorporation assay. Data represent mean ± s.e.m. from two biological replicate experiments, each performed in triplicate (*n* = 6).

## Discussion

Our work establishes *AR*-GSRs as new class of *AR* gene alteration occurring in one third of CRPC-stage tumours. The presence of *AR*-GSRs in clinical prostate cancer has not been described by previous large-scale tumour genomic profiling studies, probably because these studies have employed copy number arrays, whole-exome sequencing or moderate-coverage whole-genome sequencing, which are not powered for robust structural variant discovery in a high repeat-content gene such as *AR*[1–5]. Our work shows that *AR*-GSRs can occur in the context of *AR* amplification or a normal *AR* copy number profile. Future studies with larger sample sizes will be required to understand whether *AR*-GSRs may be mutually exclusive with *AR* mutations. Collectively, these findings indicate that the frequency of *AR* gene alterations in metastatic CRPC, which has consistently been reported at ~60–65% based on assessment of *AR* gene mutations and amplification[1,3,4], is likely to be an underestimate. Extrapolating from our finding that 3/12 (25%) of *AR* mutation- and amplification-negative CRPC metastases

harboured *AR*-GSRs indicates the *AR* gene alteration frequency in metastatic CRPC could be closer to 70–75%.

As a central tenet of precision oncology is the accurate identification of alterations in therapeutic targets such as AR, it will be important for future studies to account for the higher frequency of *AR* gene alterations presented by *AR*-GSRs. *AR* amplification or mutation has been shown to promote resistance to first-line ADT by causing aberrant AR responses to antiandrogens, castrate levels of androgens and alternative steroids[3,32–34]. Although studies of CRPC bone marrow aspirates did not find an association between *AR* amplification or mutation and resistance to abiraterone[35] or enzalutamide[36], more recent investigations have indicated that detection of increased *AR* copy number or *AR* mutations in circulating cell-free DNA is associated with resistance to these agents[11,12]. Our work shows that the presence of *AR*-GSRs at high variant allele frequency is associated with outlier, tumour-specific expression of rearrangement-dependent AR-V species that display androgen-independent, enzalutamide-resistant transcriptional activity and

can support androgen-independent growth of LNCaP cells. This is in line with studies of *AR*-GSR-positive models of CRPC, which all display outlier expression of discrete, constitutively active AR-V species, one of which is AR-V7 (refs 27,28,37). For instance, the CRPC 22Rv1 and CWR-R1 cell lines were the first models in which AR-Vs were identified and characterized functionally, and were subsequently found to be *AR*-GSR positive[15,16,18,38]. These models require high AR-V7 expression for the CRPC phenotype, as evidenced by AR-V7-specific knockdown inhibiting androgen-independent growth and restoring sensitivity to enzalutamide[19]. Other CRPC models in which alternative AR-Vs were found to be highly expressed, including ARv567es in LuCaP 86.2 and LuCaP 136 xenografts[21], and mouse AR-V2/V4 in the Myc-CaP cell line[22], also harbour underlying *AR*-GSRs[28,31] or display complex AR splicing patterns consistent with underlying *AR*-GSR events[22].

Given the link between *AR*-GSRs and high AR-V7 expression in model systems, it was an unexpected finding that *AR*-GSRs were not associated with AR-V7 expression in metastatic CRPC tissue. AR-V7 has received considerable attention as a predictive and/or treatment selection biomarker in CRPC, because detection of AR-V7 mRNA in circulating tumour cells is associated with resistance to abiraterone and enzalutamide but not taxane chemotherapy[13,14,39]. However, these results with circulating tumour cells do not seem to align with RNA-seq studies of tumour tissue, which have shown that AR-V7 mRNA is detectable in virtually all CRPC[3]. This is further confounded by data showing that ratios of AR-V7 to AR are nearly identical in CRPC versus primary prostate cancers, normal prostate tissues, breast cancer and even normal breast tissue[3,24,25]. Our finding that AR-V7 expression in CRPC is associated with *AR* gene copy number may reconcile these discrepancies. For example, whereas RNA-seq studies have assessed AR-V7/AR expression ratios[3,24,25], studies with circulating tumour cells have employed a binary yes/no readout for AR-V7 mRNA detection by RT–PCR[13,39,40]. Our data raise the possibility that AR-V7 mRNA, which is nearly universally expressed at <10% of the level of full-length AR, may be below the threshold of detection in circulating tumour cells unless *AR* gene copy number is increased. With this in mind, we speculate that studies linking *AR* copy number increases in blood to enzalutamide and abiraterone resistance[11,12], and studies linking AR-V7 mRNA detection in circulating tumour cells to enzalutamide and abiraterone resistance[13,14] may ultimately be measuring different outcomes of increased *AR* copy number. In this context, it is difficult to know whether increased AR-V7, increased full-length AR, or both, may be promoting therapeutic resistance. This is underscored by studies with a LNCaP cell line model of CRPC progression showing that enzalutamide resistance was associated with increased expression of full-length AR and AR-V7, but only full-length AR was driving tumour cell growth and survival in the presence of enzalutamide[41]. Further studies are needed to understand the relationships between *AR* gene dosage, relative and absolute expression of AR and AR-V7, and contributions to therapeutic resistance.

A challenge presented by the detection of diverse, non-recurrent *AR*-GSR events and a limitation of our study, is that we were not able to characterize the functional impact of every detected *AR*-GSR. *AR*-GSRs that remain uncharacterized were all present at low variant allele fractions and existed concurrent with *AR* gene amplification. The lack of tumour-specific AR-V expression in tumours C-14A and C-14B highlights the possibility that *AR*-GSRs present at low variant allele fraction may simply reflect bystander events resulting from the *AR* gene amplification process and/or a high burden of genomic rearrangements in CRPC[24]. An additional limitation of our study is that our pipeline

consisting of AR hybrid capture, Illumina paired-end sequencing and the use of LUMPY and Delly structural variant detection algorithms probably precluded 100% sensitivity for *AR*-GSR discovery. For example, high repeat content combined with the short reads available from Illumina sequencing prevented coverage of ∼15% of the *AR* gene. In addition, although the intersecting calls from LUMPY and Delly yielded *AR*-GSR predictions that could be validated with orthogonal approaches, numerous alternative structural variant detection algorithms are available that may have greater sensitivity in the contexts of sub-clonality and repetitive DNA[42].

It will be important in future longitudinal studies to deploy tools for detection and quantification of *AR*-GSRs, to determine whether they are associated with primary resistance to enzalutamide or abiraterone, or whether they emerge and/or become enriched during therapy. Nevertheless, our study advances understanding of the prostate cancer genome by identifying the frequency, spectrum and functional impact of widespread *AR*-GSR events in clinical prostate cancer tissues. Although the precise breakpoint locations and signatures of *AR*-GSRs are non-recurrent, the expression of COOH-terminally truncated AR-Vs with constitutive activity appears to be at least one recurrent functional outcome across prostate cancer cell lines, patient-derived xenografts and clinical tissues with high variant allele fractions of *AR*-GSRs. Collectively, these findings highlight *AR*-GSR-driven AR-Vs as important mediators of resistance and attractive therapeutic targets in prostate cancer.

## Methods

**Tissue acquisition.** Metastatic CRPC samples were obtained from patients who died of CRPC in 2000–2013 and who signed written informed consent for a rapid autopsy performed within 6 h of death, under the aegis of the Prostate Cancer Donor Program at the University of Washington[43]. Localized CRPC samples were obtained under written informed consent by TURP procedures performed for obstructive uropathy at the University of Texas Southwestern Medical Center. Hormone-naive prostate cancer tissues from patients with biopsy-confirmed prostate cancer (Gleason Score 6–9) were obtained under written informed consent at the time of radical prostatectomy from the University of Texas Southwestern Medical Center tissue core under UTSW IRB STU 112013-056. The studies with these tissues described in this study were approved by Institutional Review Boards at the University of Minnesota, University of Washington and University of Texas Southwestern Medical Center. The origin, characteristics and AR expression status of LuCaP 86.2 and LuCaP 145.1 patient-derived xenografts has been described[21,27,28].

**AR DNA sequencing.** Genomic DNA isolated from prostate cancer tissues was submitted to the University of Minnesota Genomics Center for DNA-seq library preparation and hybrid capture with a custom SureSelect (Agilent) bait library using the SureSelect QXT reagent kit (Agilent) as per the manufacturer's recommendations. The SureSelect bait library was created using the SureSelect DNA Standard Design Wizard (Agilent) with 5× tiling density, moderately stringent masking and MaximizePerformance parameters selected (Agilent). Post-capture sequencing libraries were pooled and diluted to 10 pM for flow cell clustering and sequenced in 2 lanes of an Illumina HiSeq 2500 with 2 × 150 bp settings for CRPC metastases, or 1 lane of an Illumina HiSeq 2500 with 2 × 125 bp settings for localized prostate cancer.

**AR-GSR detection.** FASTQ files of the paired-end reads were mapped with BWA-MEM (v0.7.10)[44] to human reference hg19 with default parameters. Reads with mapping quality <20 were discarded. Duplicates were removed using Picard Mark Duplicates v1.68 (http://picard.sourceforge.net). Discordantly mapped read pairs and split reads were extracted as recommended by LUMPY instructions (https://github.com/arq5x/lumpy-sv). LUMPY (v0.2.8)[45] was first used to detect structural variants using both paired and split reads by setting the mean and s.d. of insert size as 300 and 50, respectively. In addition, Delly (v0.5.9)[46] was also used to detect structural variants from processed BAM files. LUMPY and DELLY structural variant calls were merged if the two breakpoints for each structural variant were within 1 kb of each other. *AR*-GSRs were defined as structural variants supported by both LUMPY and DELLY with at least one of the breakpoints mapping within the AR gene body (RefSeq full-length transcript NM_000044.3; chrX:66763873-66950461). Candidate *AR*-GSRs with at least ten supporting paired-end reads and ten supporting split reads were selected for further validation. Tumour P-4A displayed three candidate AR-GSRs that met these criteria, but inspection of these

and other structural variant calls from P-4A revealed that all were deletions with breakpoints at known splice donor/acceptor junctions in wild-type AR mRNA, indicating this sample was contaminated with AR complementary DNA (Supplementary Data 2). This sample was therefore classified as *AR*-GSR negative. Variant allele fraction of each *AR*-GSR candidate was calculated using SHEAR[29].

**AR-GSR validation.** Data supporting each *AR*-GSR call (BAM files containing aberrantly mapping DNA-seq reads or concordantly mapping DNA-seq reads, as well as BED files representing the locations of putative breakpoints) were inspected manually in integrative genomics viewer[47]. Based on this manual interpretation, PCR primers were designed to specific genomic regions flanking the predicted break fusion junctions, with binding orientations determined by whether the *AR*-GSR was due to deletion, duplication, inversion or trans-location. PCR primers are listed in Supplementary Table 7. Genomic DNA from CRPC tumours was subjected to whole-genome amplification using the REPLI-g Amplification kit (Qiagen) and purified using a QIAquick Nucleotide Removal Kit (Qiagen). Whole-genome amplification DNA was used for PCR with specific primer pairs using AccuStart II PCR SuperMix (Quanta Biosciences) as per the manufacturer's recommendations. PCR products of the expected size were isolated from gels using a QIAquick Gel Extraction Kit (Qiagen) and subjected to direct Sanger sequencing using the forward or reverse PCR primer, or alternatively cloned using the TOPO PCR cloning system (Thermo Fisher) and subjected to Sanger sequencing using universal M13 forward and reverse primers (M13F: 5′-GTAAAACGACGGCCAGT-3′ and M13R: 5′-CAGGAAACAGCTAT GAC-3′).

**Determination of AR gene copy number.** *AR* gene (RefSeq transcript NM_000044.3; chrX: 66763873-66950461) copy number was determined for each tumour specimen by calculating the ratio of normalized mean AR read coverage of a tumour sample versus normalized mean AR read coverage for an AR copy number alteration negative tumour sample (C-10A). Mean AR read coverage was normalized based on five control regions included on the SureSelect capture panel: C1 (chr9:98258995-98264381), C2 (chrX:16153017-16159789), C3 (chr14:105249980-105255049), C4 (chr15:40514992-40520036) and C5 (chr15:67390102-67395049). The average mean coverage of each region was calculated by BedTools[48]. Normalized AR coverage was calculated by the following equation 1:

$$\text{Ratio} = \text{mean\_read\_coverage of AR/Median}$$
$$\text{of } [\text{mean\_read\_coverage (C1)}/2, \text{mean\_read\_coverage (C2)},$$
$$\text{mean\_read\_coverage(C3)}/2, \text{mean\_read\_coverage(C4)}/2,$$
$$\text{mean\_read\_coverage(C5)}/2]$$

**Determination of AR single-nucleotide variants.** *AR* sequence variants were called using FreeBayes v0.9.14-14-gb00b735 (ref. 49) and VarScan 2 v.2.3.6 (ref. 50). Variants were called on BAM files generated for *AR*-GSR detection. Using FreeBayes, variants were called using the pooled-discrete and genotype-qualities settings. To be called, variants had to be represented by at least three reads and 10% of the total reads. The resulting variants were then filtered to retain those with a minimum total depth of 20. Variants were also called using Varscan2; after filtering reads for a minimum mapping quality of 5, variants had to reach at least 1% of the total reads in order to be called. Sequence variants files were merged with bcftools v. 1.0 (ref. 51). Sequence variants called by both FreeBayes and VarScan 2 were considered *bona fide* AR mutations.

**RNA isolation and quantitative RT–PCR.** Total RNA was isolated from sections of frozen tissue using RNA STAT-60 (Tel-Test), while incorporating Phase Lock Gel Heavy tubes (5 Prime) into the protocol, to better facilitate phase separation. RNA yield and purity was obtained on a NanoDrop 2000 (Thermo Fisher Scientific). RNA Integrity was evaluated on a 2100 Bioanalyzer (Agilent Technologies). Total RNA was subjected to RT–PCR using TaqMan primers and probes designed to specifically amplify and interrogate the exon 4/5 splice junction (full-length AR, ThermoFisher Hs00171172_m1), the exon 3/CE3 splice junction (AR-V7; Supplementary Table 8) and the exon 4/8 splice junction (ARv567es, Supplementary Table 8). Data were transformed using the differential Ct method relative to RPL13A as control (ThermoFisher Hs04194366_g1) as described[52]. Assays were performed using TaqMan reagents (ThermoFisher) as per the manufacturer's recommendations.

**RNA sequencing.** For RNA-seq experiments, RNA was subjected to Illumina-based RNA-seq using a TruSeq Stranded mRNA LT Sample Prep Kit (Illumina) as per the manufacturer's recommendations. RNA-seq libraries were sequenced with an Illumina HiSeq 2500 with $2 \times 50$ bp settings, resulting in $70$–$100 \times 10^6$ reads per sample.

**Integrative analysis of RNA-seq and AR-GSR data.** Tumour-specific reference genome assemblies were developed by adding a derivative chromosome to the hg19 reference genome. This derivative chromosome represented a rearranged *AR* architecture design based on the exact break fusion junction signature(s) arising from a single *AR*-GSR event. To prevent errors from reads mapping to loci represented in both the derivative chromosome and hg19 chromosomes, the endogenous *AR* locus (chrX) and other appropriate endogenous sequences were masked. RNA-seq reads from selected tumours were mapped to these 'tumour-specific' reference genome assemblies using HISAT2 (ref. 53) or TopHat2 (ref. 54). Bins were created for each exon and intron within the derivative chromosome assembly and heatmaps were generated in R using HISAT2 data to provide a quantitative readout of the number of split reads spanning these specific exon/exon, exon/intron or intron/intron bins. These heatmaps were used to highlight the locations of canonical and aberrant AR splicing events. Genomic locations on derivative chromosomes that corresponded to bins with unexpected, but quantitatively important splice junctions were inspected manually in integrative genomics viewer[47] using BAM files of mapped RNA-seq reads as well as splice junction bed files from HISAT2 and TopHat2. Splice junction bed files were generated by filtering HISAT2 bams to retrieve split reads, which were then converted to bed files with Bedtools and filtered to retain only those split reads where at least 5 bp mapped to each side of the junction. Mapping RNA-seq reads from different tumours of the same subject served as a control, to ensure novel read-mapping patterns and rearrangement-dependent splice junctions were occurring specifically in the context of the relevant tumour-specific *AR*-GSR. Comrad[30] was used with default settings, to detect AR fusion transcripts from RNA-seq and AR DNA-seq data from tumours C-9A, C-9B, C-12A, C-12B, C-14A and C-14B.

**ARv567es immunohistochemistry.** Immunohistochemistry experiments were performed as described[52]. Briefly, formalin-fixed paraffin-embedded tissue sections (5 μm) from subject C-6 and patient-derived xenograft tissues expressing ARv567es (LuCaP 86.2) or lacking AR expression (LuCaP 145.1) were deparaffinized and rehydrated. Antigen retrieval was performed with 10 mM citrate buffer (pH 6.0) in a pressure cooker (20 psi for 10 min). Endogenous peroxide and biotin/avidin was blocked for 15 min with respective agents (Vector Laboratories). After incubating with 5% normal goat–horse–chicken serum at room temperature for 1 h, sections were incubated with ARv567es mAb EP345 (Epitomics/Abcam) at 1:200 dilution at 4 °C overnight, followed by biotinylated secondary antibody and sABC reagent (Vector Laboratories). DAB (Invitrogen) was used as the chromogen and haematoxylin as the counterstain.

**Plasmids.** Plasmid constructs harbouring full-length AR-V7 (AR 1/2/3/CE3), ARv567es (AR 1/2/3/4/8), -5746 PSA-LUC and 4XARE-E4-LUC have been described[31]. Plasmids encoding AR 1/2/3/4-ups/5a-ups, AR 1/2/3/4-ups/5b-ups, AR 1/2/3/4-ups/5c-ups and AR 1/2/3/4-chr11 were generated using a previously described strategy[15]. Briefly, synthetic cassettes encoding COOH-terminal extension sequences (Supplementary Table 9) were annealed, phosphorylated and ligated into an XbaI-mutant version of the p5HBhAR-A expression plasmid[15] prepared by digestion with XbaI. All constructs were verified by DNA sequencing. To construct lentiviral expression vectors, the AR-V coding sequences were liberated from these p5HBhAR-A-based plasmids using PCR with primers designed to incorporate 5′-EcoRV and 3′-SalI sites (forward primer: 5′-TGGGAT ATCCAGCCAAGCTCAAGG-3′ and reverse primer: EcoRV/SalI-digested PCR products were then ligated with EcoRV/SalI-digested pLV-GFP[31] to replace the GFP insert with AR-V cDNAs. Lentivirus constructs were verified by DNA-seq, restriction mapping, transient transfection and western blotting with an antibody specific for the AR amino terminus.

**Cell lines.** LNCaP (ATCC, CRL-1740), VCaP (ATCC, CRL-2876) and DU145 (ATCC, HTB-81) cells were obtained from American Type Culture Collection (ATCC) and cultured according to ATCC protocol. ATCC ensures authenticity of these human cell lines using short tandem repeat analysis. LNCaP and DU145 cells were maintained in RMPI 1640 (Invitrogen) with 10% fetal bovine serum (FBS), 100 units per ml penicillin and $100 \,\mu\text{g ml}^{-1}$ streptomycin (pen/strep) in a 5% $CO_2$ incubator at 37 °C. VCaP cells were cultured in DMEM medium (Invitrogen) with 10% FBS and pen/strep in a 5% $CO_2$ incubator at 37 °C. Aliquots of cell culture supernatants from cells in active culture were evaluated regularly for mycoplasma contamination using a PCR-based method as described[55]. All experiments with LNCaP, VCaP and DU145 cells were performed within 2–3 months of resuscitation of frozen cell stocks prepared within three passages of receipt from ATCC.

**Luciferase reporter gene assays.** LNCaP cells were transfected by electropora-tion exactly as described[15]. Briefly, 12 μg of DNA (9 μg 5746 PSA-LUC, 3 μg SV40-*Renilla* and 0.42 μg AR-V expression plasmid) were mixed with $4 \times 10^6$ LNCaP cells in a 4 mm gap-width electroporation cuvette and subjected to a 305 V, 10 ms pulse using a Square Wave Electroporator (BTX/Harvard Apparatus) before plating in RPMI medium supplemented with 10% charcoal-stripped, steroid-free FBS (CSS). VCaP and DU145 cells were transfected using Superfect (Qiagen) as per

the manufacturer's recommendations exactly as described[31]. Briefly, 1.2 μg of 4XARE-E4-LUC, 0.4 μg SV40-REN, 40 ng of AR-V expression plasmid and 8 μl Superfect reagent were suspended in 68 μl of serum-free DMEM, incubated for 20 min, diluted with 480 μl of DMEM supplemented with 10% CSS and added to 1 well of a 24-well plate that had been seeded with $1 \times 10^5$ cells the day prior. For all reporter-based experiments, 24 h post transfection, cells were re-fed with serum-free medium containing 1 nM dihydrotestosterone (Sigma) or 0.1% ethanol as vehicle control in combination with 30 μM enzalutamide (Selleck Chemical) or 0.1% DMSO as vehicle control for 24 h. Cells were harvested in $1 \times$ passive lysis buffer provided in a Dual Luciferase Assay Kit (Promega). Activities of the firefly and *Renilla* luciferase reporters were assayed using a Dual Luciferase Assay Kit as per the manufacturer's recommendations. Transfection efficiency was normalized by dividing firefly luciferase activity by *Renilla* luciferase activity. Data presented represent the mean ± s.e.m. from three independent experiments, each performed in duplicate.

**Lentivirus infection and cell proliferation assays.** Lentivirus encoding GFP, AR 1/2/3/4-ups/5a-ups, AR 1/2/3/4-ups/5b-ups, AR 1/2/3/4-ups/5c-ups and AR 1/2/3/4-chr11 was prepared using a standard third-generation packaging system in 293T cells. Briefly, 293T cells were co-transfected with lentivirus vectors and packaging vectors pCMVΔR8.91 and pMD.G (ref. 31) at a ratio of 4:3:1 using Lipofectamine 2000 (ThermoFisher Scientific). Medium containing lentivirus was collected from 48 to 96 h post transfection, pooled and concentrated using Lenti-X Concentrator (Clontech). LNCaP cells were seeded in 6 cm dishes at $4 \times 10^5$ cells per dish in RPMI with 10% FBS and pen/strep. Cells were transduced the next day by addition of 0.5, 2, 8 or 32 μl of lentivirus directly to tissue culture medium. Cells were re-seeded 120 h post transduction on 96-well plates in RPMI + 10% CSS + pen/strep, and subjected to cell proliferation assays using a BrdU ELISA kit (Roche) according to the manufacturer's recommendations.

**Western blotting.** Transfected cells were harvested in $1 \times$ Laemmli buffer and subjected to western blot analysis as described[15]. Blots were incubated with primary antibodies (AR-N20, Santa Cruz and ERK2-D2, Santa Cruz) diluted 1:1,000 overnight at 4 °C and then incubated with the appropriate horseradish peroxidase-conjugated secondary antibodies diluted 1:10,000 at room temperature for 2 h. Blots were developed by incubation with Super Signal chemiluminescence reagent (Pierce) and exposed to film. Uncropped films are shown in Supplementary Fig. 14.

**Statistics.** Tests for higher expression of AR or AR-V7 mRNA in tissues positive for *AR*-GSRs or copy number increase were performed using one-sided Mann–Whitney U-tests. Mann–Whitney *U*-tests were selected based on sample distributions being non-normal in Shapiro–Wilks goodness-of-fit tests. Sample sizes were not chosen to ensure adequate power, to determine a pre-determined effect size. Statistical calculations were performed using GraphPad Prism 5 software.

**Data availability.** The RNA-seq and AR DNA-seq data that support this study are available through the NCBI database of Genotypes and Phenotypes (dbGaP) under accession phs001223.v1.p1. All other relevant data are available within the article file or Supplementary Information or available from the authors upon request.

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

## Acknowledgements

We thank the patients and their families who were willing to participate in the Prostate Cancer Donor Program. We acknowledge Drs Robert Vessella, Bruce Montgomery, Evan Yu, Elahe Mostaghel, Heather Cheng, Paul Lange, Martine Roudier and Lawrence True, and the tissue acquisition teams for their contributions to the University of Washington Prostate Cancer Donor Program. In addition, we thank Dr Eva Corey for access to LuCaP xenograft tissue and Belinda Nghiem for her technical expertise. This research was supported by funding from NIH grants R01CA174777 (to S.M.D.), U.S. Department of Defense Prostate Cancer Research Program Transformative Impact Award W81XWH-15-1-0430 (to S.R.P. and S.M.D.), the Pacific Northwest Prostate Cancer SPORE (P50CA97186 to P.S.N.), PO1 NIH grants (PO1CA085859 and PO1CA163227) and the Richard M. Lucas Foundation. T.M. was supported by NIH Medical Scientist Training Program grant T32GM008244.

## Author contributions

C.H., Y.L., R.Y. and S.M.D. conceived the study and designed the methodologies and experiments. Y.L., T.M., Y.H., C.S., G.L., I.C. and S.M.D. performed experiments and acquired data. C.H., Y.L., R.Y., T.M., Y.H., C.S., G.L., I.C., R.L., V.K., T.H.H., S.R.L., P.S.N., S.R.P. and S.M.D. analysed and interpreted the data. B.L., G.V.R., C.S.H. and C.M. acquired and processed tissues for analysis. C.H., P.S.N., S.R.P. and S.M.D. prepared the manuscript.
