## [Peer Review File · Nature Communications]

Reviewers' comments:

Reviewer #1 (Remarks to the Author): Expert in androgen receptor signalling

First line treatment for advanced prostate cancer (PCa) is androgen deprivation therapy (ADT), which is often durably effective. However, the transition to castration-resistant PCa (CRPC) invariably occurs, and androgen receptor (AR) activity is often restored despite continued therapeutic targeting, and in fact it is rewired. With the recent approval of abiraterone and enzalutamide (ENZ), patient survival has increased, yet the efficacy of these agents is, unfortunately, also transient. As such, developing additional means to thwart AR activity in PCa is of great clinical importance. One mechanism of resistance to anti-androgen therapy is selection of AR splice variant (AR-Vs) species that lack the ligand binding domain (LBD), rendering AR constitutively active. The present study seeks to determine the relevance of AR gene rearrangements in clinical specimens. The rationale for this study is solid, and the authors assessed metastatic and localized CRPC, as well as ADT-naïve localized PCa.

Herein, the authors determined that 12/30 mCRPC tumors harbored AR gene amplification, 6/30 harbor missense mutations, and 10/30 harbor AR genomic structural rearrangements (AR-GSRs). These GSRs occurred in both AR amplified and non-amplified tumors, but not in those harboring missense mutations. AR-GSRs were also found present in localized CRPC, and it was determined that AR-GSRs are more frequent in CRPC than in ADT-naïve PCa. The AR-GSR breakpoints were found to be unique to each patient, harbor NHEJ signatures, and were frequently in LINEs or SINEs. In order to determine the impact of the AR-GSRs, AR-V7 levels were compared to AR-GSR positivity and AR copy number, and it was determined that AR-V7 was correlated with AR copy number, not the presence of AR-GSRs. This prompted a candidate-by-candidate approach which was used to identify the functional significance of AR-GSRs on AR-V expression. Using this approach, it was determined in the CRPC metastases of one patient that ARv567es expression was associated with a clonal complex AR-GSR. Furthermore, in a different tumor tandem duplication was enriched and was associated with several novel AR-Vs that harbor constitutive transcriptional function. Finally, in a third tumor, AR translocation was found to be associated with the expression of another novel AR-V that harbors constitutive transcriptional activity.

There is a clinical need to identify which patients may or may not respond to AR-directed therapeutic strategies in the name of precision medicine. This study indicates that up to 75% of CRPC may harbor AR-GSRs, which are associated with constitutively active AR-V expression. While the study is justified, well-designed, and well-executed, several suggestions for increasing the impact and enthusiasm are outlined below:

1. It should be determined if all AR-GSR events are associated with AR-V expression, or merely a subset.
2. While the luciferase assays in Figures 4G and 5G are interesting, confidence in the stated conclusions would benefit from confirmation using other systems/techniques. In addition to caveats of overexpression, the cell lines used express a mutant copy of AR, and the authors have indicated that AR-GSRs are likely mutually exclusive to missense mutations.

Reviewer #2 (Remarks to the Author): Expert in clonal evolution

The manuscript by Henzler, Li, Yang et al. shows that structural variants in AR are frequent in castration resistant prostate cancer (CRPC). The study uses material from autopsies from 15 patients that died of CRPC (as well as a small TURP and prostatectomy series for reference) and performs targeted sequencing of the AR locus. The authors show the presence of somatic structural variants in 6 of the 15 CRPC patients, and perform a detailed characterization of these

variants and the resulting AR variant products at the RNA and protein level, for three of the cases. This represents a reasonably small, but well focussed study of AR rearrangements that goes into considerable depth. Because of the limited size of the study, the exact frequency of such AR structural variants in CRPC is hard to assess, but it is already clear from this study that these should not be ignored and considered/assayed alongside AR amplifications and point mutations in AR. As such, the study advances our understanding of CRPC and may down the line impact therapy for a subset of patients.

Specific comments:

- Assessing clonality based on structural variants is difficult, but the author's analysis is sound. At present, they cannot determine if low variant allele fractions of some structural variants are due to low purity of the samples or subclonality of the variants. Copy number analysis from SNP arrays of the samples could potentially help with that. They may also improve the copy number calling of the AR locus.
- Minor remark regarding materials and methods: I would not call the derivative chromosomes they add to the reference genome 'fictitious' as this derivative chromosome was actually observed.

Reviewer #3 (Remarks to the Author): Expert in cancer genomics and structural variants

This study provides data supporting that genomic structural rearrangements in androgen receptor (AR-GSRs) form a novel class of molecular alterations that occur in CR Prostate Cancer. The authors also state that there is substantial heterogeneity in AR-GSR break-point location in the CRPC tumors. Some of these breakpoints are proposed to be responsible for driving synthesis of AR-variant species lacking the ligand binding domain (LBD). This is an intriguing observation providing early insight into mechanisms involving GSRs that activate AR-independent pathways in CRPC. Overall, this is an interesting study for the cancer research community.

Having said that, I have number of questions and comments especially related to the computational methodology that need to be addressed. In particular I am surprised that the authors used Lumpy, a program which is not commonly used for cancer SV detection purposes. Lumpy is not designed to identify SVs in repetitive regions of the human genome. In addition, it is not aimed to resolve SVs in clonal samples. I would test a number of methods (Delly again is not designed for repetitive DNA) before I reach any conclusions. Breakdancer, VariationHunter, CommonLAW and Pindel are other SV callers that can be used for this purpose. It is unfortunate that none of these methods are designed to handle clonal variation; additionally they are typically not good in detecting micro structural arrangements.

Perhaps collectively (by evaluating the whole collection of SVs they predict rather than considering only the intersecting calls) they will be able to detect crucial signals.

An additional issue I would like to raise is in integrative analysis of RNA-seq and genomic structural rearrangements. There are good tools developed specifically for that purpose, such as Comrad and nFuse. Although, in principle, these tools are used for identifying transcribed fusions between two distinct genes, there is no reason why they shouldn't be used here for the authors' purposes.

Additional comments follow.

1. Although, GSRs within AR are understudied and not commonly reported, it was not very surprising that several AR-GSRs were detected in CRPC as opposed to hormone-naïve prostate cancer. Many recent studies have shown high burden of complex genome rearrangements in advanced CRPC cases and these GSRs are typically not driver events.

2. Were there any splice-site mutations within AR, especially in the AR-LBD mutational hotspot region?

3. The authors have concluded that AR-GSRs can co-occur with AR-amplification or normal AR copy number profile, but seem to be mutually exclusive with AR mutation:

a. the authors should provide a statistical assessment of mutual exclusivity or co-occurrence to support their claim;

b. is there a high burden of AR-GSR in regions where with no AR-amplification or mutations in AR?

4. Although, the author's claim of AR-GSR sub-clonality in CRPC-metastatic tumors (page-7) is supported by the data presented, the claim that these tumors represent cell fractions expressing high levels of AR-Vs is not very convincing.

5. The authors have associated AR-GSR with outlier expression of ARv567es that lack LBD. The nomination of ARv567es seems very biased as there are many other known AR-Vs that lack LBD. I would be better convinced if the authors could show the AR-isoform expression quantification in RNAseq data.

Response to Reviewer #1, an expert in androgen receptor signaling

Comment 1

It should be determined if all AR-GSR events are associated with AR-V expression, or merely a subset.

Response 1

Not all AR-GSR events are associated with AR-V expression. To clarify this concept, we made three revisions to the manuscript. First, we added a new sub-section to the end of the Results entitled “Not all Detected AR-GSRs Underlie Expression of Tumor-Specific AR-Vs”. In this sub-section, we highlighted a set of low variant allele fraction AR-GSRs identified in tumors 00-169C, 04-050P, and 04-050S that would be incompatible with expression of AR-Vs as well as full-length AR. Second, we carried out integrative analysis of RNA-seq and AR DNA-seq data from tumors 13-042M and 13-042O. These tumors harbored AR-GSRs at low variant allele fraction (1-5%). There was no evidence for expression of rearrangement-specific AR-Vs in either of these tumors (new Supplementary Data 3). Third, we edited the third paragraph of the Discussion to elaborate on the notion that low variant allele fraction AR-GSR events may be bystander effects associated with AR gene amplification and/or a high degree of genomic instability in CRPC. Additional edits were made to this paragraph to clarify the main conclusion that AR-GSRs at high variant allele fraction are associated with expression of known as well as novel AR-Vs.

Comment 2

While the luciferase assays in Figures 4G and 5G are interesting, confidence in the stated conclusions would benefit from confirmation using other systems/techniques. In addition to caveats of overexpression, the cell lines used express a mutant copy of AR, and the authors have indicated that AR-GSRs are likely mutually exclusive to missense mutations.

Response 2

This is an excellent point. We therefore added luciferase data from two additional systems: the DU145 cell line (no full-length AR expressed) and the VCaP cell line (AR-amplified). These cell lines better reflect the contexts in which AR-GSRs were observed in the clinical specimens. Overall, our results are consistent with the assays in LNCaP cells, demonstrating that the novel AR-Vs discovered in tumors 12-005H and 05-217F display constitutive, androgen-independent transcriptional activity. These new data are shown in Supplementary Figures 10 and 12 of the revised manuscript. Additionally, we generated lentiviral vectors for expression of these AR-Vs in LNCaP cells, and demonstrated they all had the ability to promote androgen-independent growth compared to expression of GFP. These new data are shown in Figs. 4h and 5h of the revised manuscript.

Response to Reviewer #2, an expert in clonal evolution

Comment 1

Assessing clonality based on structural variants is difficult, but the author's analysis is sound. At present, they cannot determine if low variant allele fractions of some structural variants are due to low purity of the samples or subclonality of the variants. Copy number analysis from SNP arrays of the samples could potentially help with that. They may also improve the copy number calling of the AR locus.

Response 1

When CRPC tissue sections were cut for DNA isolation, adjacent sections were cut to determine % cancer (based on H&E staining). We added a column to Supplementary Table 2 to indicate the % cancer cells present in these adjacent tissue sections. We also edited text in the Results section accordingly. With the exception of tumor 12-01118, all samples were high purity. Therefore, the low variant allele fractions observed for AR-GSRs and AR somatic mutations were due to sub-clonality.

Comment 2

Minor remark regarding materials and methods: I would not call the derivative chromosomes they add to the reference genome 'fictitious' as this derivative chromosome was actually observed.

Response 2

We replaced all instances of the word "fictitious" with "derivative" in this paragraph of the materials and methods section.

Response to Reviewer #3, an expert in cancer genomics and structural variants

Comment 1

I have number of questions and comments especially related to the computational methodology that need to be addressed. In particular I am surprised that the authors used Lumpy, a program which is not commonly used for cancer SV detection purposes. Lumpy is not designed to identify SVs in repetitive regions of the human genome. In addition, it is not aimed to resolve SVs in clonal samples. I would test a number of methods (Delly again is not designed for repetitive DNA) before I reach any conclusions. Breakdancer, VariationHunter, CommonLAW and Pindel are other SV callers that can be used for this purpose. It is unfortunate that none of these methods are designed to handle clonal variation; additionally they are typically not good in detecting micro structural arrangements.

Response 1

We agree with the reviewer that each structural variant detection algorithm brings trade-offs regarding sensitivity and specificity, especially for cancer SV detection, sub-clonality, and repetitive regions of the genome. The challenges with high repeat content in the AR locus are further confounded by the use of a SureSelect hybrid-based capture panel, which has highly repetitive regions of AR masked (as outlined in the first paragraph of the Results section of p4 and summarized in Supplementary Figure 1 and Supplementary Tables 1 and 4). As the goal of the study was to establish whether AR-GSRs occurred in clinical CRPC, we focused our analysis on high-confidence calls rather than developing computational methods that would return exhaustive lists of candidate SVs. This is also why we spent many months validating the calls from our SV detection pipeline using PCR and Sanger sequencing. In each case, the SV calls we prioritized for follow-up (i.e. at least 10 split reads and 10 paired-end reads supporting the intersecting SV calls from LUMPY and Delly outputs) could be validated using this orthogonal approach. This clearly demonstrates that the computational methodology we employed had high specificity. We have been studying AR genomic structural variation for several years now, and have tested (and published) use of several SV callers, beginning with Hydra as outlined in our publication in *Oncogene* 2012 Nov 8;31(45):4759-67. We have since moved to use of LUMPY and Delly because these algorithms require both paired-end and split-read support for SV calling. Further, Delly is the SV detection algorithm used in the MSKCC clinical cancer pipeline (MSK-IMPACT: Cheng DT et al., *The Journal of Molecular Diagnostics*, Vol. 17, No. 3, May 2015). We would like to point out that the output from our LUMPY/Delly computational methodology provided many additional calls in which we had less confidence for follow-up (i.e. less than 10 split reads and 10 paired-end reads supporting the SV call, see Supplementary Data 2).

Nevertheless, to directly respond to the reviewers' concern, we implemented Pindel as an alternative SV caller using our AR DNA-seq data as input. Overall, Pindel returned many more calls than our LUMPY/Delly strategy (summarized in the attached spreadsheet for review purposes). Interestingly, there was very little overlap between the output from Pindel and the intersecting LUMPY/Delly calls we had reported in Supplementary Data 2. Closer inspection of these LUMPY/Delly and Pindel overlapping calls revealed that Pindel "missed" the translocation that had been validated by PCR and Sanger

sequencing in tumor 05-217F, and which resulted in the novel AR-V discovered in our study (Fig. 5). Pindel also missed several additional AR-GSRs that we had validated via PCR and Sanger sequencing (summarized in the attached spreadsheet for review purposes). Collectively, we feel these considerations provide considerable support for our focusing on intersecting calls from LUMPY and Delly.

Comment 2

Perhaps collectively (by evaluating the whole collection of SVs they predict rather than considering only the intersecting calls) they will be able to detect crucial signals.

Response 2

We were able to detect critical signals as demonstrated by our successful ability to validate LUMPY/Delly intersecting calls using the orthogonal approach of PCR/Sanger sequencing. As outlined in our response to Comment #1 above and highlighted in Supplementary Data 2, our main challenge was detection specificity, not detection sensitivity. If we had considered the entire collection of SV calls from LUMPY, Delly, and Pindel, we would have been dealing with hundreds of candidate SVs in each tumor sample! We do not think this would have been a tenable starting point for the study.

Comment 3

An additional issue I would like to raise is in integrative analysis of RNA-seq and genomic structural rearrangements. There are good tools developed specifically for that purpose, such as Comrad and nFuse. Although, in principle, these tools are used for identifying transcribed fusions between two distinct genes, there is no reason why they shouldn't be used here for the authors' purposes.

Response 3

As suggested by the reviewer, we implemented Comrad and nFuse using RNA-seq and DNA-seq data from tumors 05-217, 12-005, and 13-047 as input. Comrad detected the AR-GSR breakpoints we had reported in tumors 12-005H and 05-217F, as well as the splicing events occurring between AR exon 3 and upstream exons (in the case of 12-005H) or chromosome 11 (in the case of 05-217F). nFuse did not identify these reported AR mRNA splicing abnormalities. We speculate this is because the nFuse algorithm was designed to identify aberrant transcripts resulting from higher-complexity genomic abnormalities such as closed chain rearrangements. The Comrad data are included as Supplementary Data 3 in the revised manuscript and are discussed in three places within the Results section. The appropriate reference has also been added (McPherson et al., Bioinformatics, 2011). The nFuse data are included for reviewer purposes. We thank the reviewer for suggesting these algorithms, as implementation of Comrad provided validation of the novel AR splice junctions detected by our integrative analysis of RNA-seq and AR-GSRs. Nevertheless, had we relied exclusively on Comrad for our

analysis, a significant amount of manual follow-up would still have been required, using strategies similar to the integrated DNA-seq/RNA-seq analysis workflow we developed. For example, for tumor 12-005H, Comrad only reported the fusion splice junctions between AR exon 3 and upstream exons, but did not identify the splice junctions occurring between AR upstream exons as we had outlined in Figs 4b-d.

Comment 4

Although, GSRs within AR are understudied and not commonly reported, it was not very surprising that several AR-GSRs were detected in CRPC as opposed to hormone-naïve prostate cancer. Many recent studies have shown high burden of complex genome rearrangements in advanced CRPC cases and these GSRs are typically not driver events.

Response 4

We agree with the reviewer. As outlined in our response to Reviewer 1, Comment 1, integrative analysis of DNA-seq and RNA-seq data from AR-GSR positive tumors 13-042M and 13-042O did not reveal tumor-specific AR-V splicing patterns. Thus, not all AR-GSR events are associated with AR-Vs. The main distinction between 13-042M/O and the tumors that displayed AR-GSR-driven AR-Vs is the degree of clonal enrichment (13-042M and O displayed AR-GSRs in the 1-5% variant allele fraction range as outlined in Table 1). This further highlights the importance of using quantitative approaches like our SHEAR algorithm when evaluating AR-GSRs. We added a new Supplementary Data 3 and a corresponding paragraph to the Results to reflect these findings with 13-042M/O. We also expanded the Discussion on this topic to further clarify the possibility that certain AR-GSR events may be bystanders. We hope these revisions provide more clarity on the differences observed between AR-GSRs at high clonal fraction vs. AR-GSRs present at low clonal fraction.

Comment 5

Were there any splice-site mutations within AR, especially in the AR-LBD mutational hotspot region?

Response 5

As outlined in Supplementary Table 4, our AR DNA-seq approach provided deep coverage of the majority of the coding and non-coding regions of the AR gene. Based on our analysis of these data, we added the following sentence to the Results section of the revised manuscript: “No mutations were detected in splice donor or acceptor sites of canonical AR exons or cryptic exon (CE) 3, the 3’ terminal exon spliced in AR-V7 mRNA.”

Comment 6

The authors have concluded that AR-GSRs can co-occur with AR-amplification or normal AR copy number profile, but seem to be mutually exclusive with AR mutation. The authors should provide a statistical assessment of mutual exclusivity or co-occurrence to support their claim;

Response 6

We used Fisher's exact tests to assess mutual exclusivity and co-occurrence between AR-GSRs and AR amplification as well as AR-GSRs and AR somatic mutations. This is the same statistical approach used to test for mutual exclusivity and co-occurrence in cBioPortal, as well as a previous prostate cancer whole exome sequencing study by Golub, Meyerson, Lander, Getz, Rubin, Garraway and colleagues (Barbieri C et al, Nature Genetics, 2012). The results of these statistical tests are included in the Results section of the revised manuscript.

Comment 7

The authors have concluded that AR-GSRs can co-occur with AR-amplification or normal AR copy number profile, but seem to be mutually exclusive with AR mutation. Is there a high burden of AR-GSR in regions where with no AR-amplification or mutations in AR?

Response 7

As demonstrated by the oncoprint in Fig. 1A, there were 3 AR-GSR positive tumors (01-120B, 01-120C, and 05-217F) that lacked mutations or copy number alterations. Conversely, there were 7 AR-GSR positive tumors that were also AR amplification or mutation positive. Using Fisher's exact test, this provided a P-value of 0.7148 (3/12 vs. 7/18). Thus, there does not appear to be a high burden of AR-GSRs in tumors unaffected by other known AR alterations (mutation, amplification). Furthermore, as demonstrated in Fig. 1B, The AR-GSR breakpoints occurred in non-coding regions of the AR gene. Conversely, AR somatic mutations occurred in the coding exons of the AR gene. Therefore, there is indeed a high burden of AR-GSRs in regions with no AR mutations. Because AR amplification encompasses the entire AR gene locus (coding + non-coding), all AR-GSR breakpoints occurred in regions of the AR gene affected by amplification.

Comment 8

Although, the author's claim of AR-GSR sub-clonality in CRPC-metastatic tumors (page-7) is supported by the data presented, the claim that these tumors represent cell fractions expressing high levels of AR-Vs is not very convincing.

Response 8

The claim on p7 that AR-GSR-positive cells represent the tumor cell fractions expressing high levels of AR-Vs was based on previous work in CRPC cell lines and xenograft models as cited in references 19, 27, and 28. In the present study with clinical specimens, the tumors that we prioritized for follow-up (01-120B, 01-120C, 05-217F, and 12-005H) displayed mRNA and/or protein expression of AR-Vs that could only arise from the specific genomic configurations observed in these tumors. This is best exemplified by the AR-V detected in tumor 05-217F, which could not be synthesized in a normal cell lacking the AR:chr11 translocation. However, as outlined in our response to Reviewer 1, Comment 1, we have clarified that every AR-GSR event reported in our study is not necessarily associated with AR-V expression.

Comment 9

The authors have associated AR-GSR with outlier expression of ARv567es that lack LBD. The nomination of ARv567es seems very biased as there are many other known AR-Vs that lack LBD. I would be better convinced if the authors could show the AR-isoform expression quantification in RNAseq data.

Response 9

We agree with the reviewer that the nomination of ARv567es was biased. The study was initiated by conducting highly-specific TaqMan-based RT-PCR assays for AR-V7, ARv567es, and full-length AR in the entire cohort of CRPC metastases that had been subjected to AR DNA-seq. The rationale was that these AR-Vs had been detected in clinical prostate cancer specimens by multiple labs. The initial goal of the study was to understand if AR gene-level events correlated with expression of these known AR-Vs. This was also based on our previous work with patient-derived xenografts and genomic editing, where we proved a cause:effect relationship between AR-GSRs affecting AR exons 5-7 and expression of ARv567es (Proc Natl Acad Sci U S A. 2013 Oct 22;110(43):17492-7). To respond directly to this point, we edited this part of the Results section to change the way we presented the findings from subject 01-120. We indicated that we analyzed ARv567es mRNA because it had been linked in previous studies to AR-GSR events, and then moved into the characterization of the underlying AR-GSR.

In the conduct of our studies, we discovered a breadth of AR-GSRs at levels of high clonal enrichment that we had not anticipated. Therefore, we performed RNA-seq on select tumors to understand the impact of these AR-GSRs on AR expression patterns (we only performed RNA-seq with each of the two tumors from patients 05-217, 12-005, and 13-042). We did not perform this RNA-seq with tumors from patient 01-120. We feel that the extensive IHC-based validation of ARv567es protein expression in multiple tumors from subject 01-120 definitively links DNA-, RNA-, and protein-level events in this patient.

REVIEWERS' COMMENTS:

Reviewer #1 (Remarks to the Author):

The authors have addressed the major concerns of the study, which is much improved. The findings will be of high impact for the nuclear receptor and prostate cancer fields.

Reviewer #2 (Remarks to the Author):

The manuscript by Henzler, Li, Yang et al., focussing on AR structural variants in castration resistant prostate cancer, has been extensively revised in response to the comments from three reviewers. The new analyses are sound and I am happy to recommend publication in Nature Communications.

I agree with the author's approach of focussing on specificity rather than sensitivity of structural variant calls; the variants they report are high-confidence and well validated and are a good starting point of this detailed study.

Reviewer #3 (Remarks to the Author):

The authors have responded to my comments and questions satisfactorily with the exception of the following:

Response 1

The challenges with high repeat content in the AR locus are further confounded by the use of a SureSelect hybrid-based capture panel, which has highly repetitive regions of AR masked (as outlined in the first paragraph of the Results section of p4 and summarized in Supplementary Figure 1 and Supplementary Tables 1 and 4). As the goal of the study was to establish whether AR-GSRs occurred in clinical CRPC, we focused our analysis on high-confidence calls rather than developing computational methods that would return exhaustive lists of candidate SVs. This is also why we spent many months validating the calls from our SV detection pipeline using PCR and Sanger sequencing. In each case, the SV calls we prioritized for follow-up (i.e. at least 10 split reads and 10 paired-end reads supporting the intersecting SV calls from LUMPY and Delly outputs) could be validated using this orthogonal approach. This clearly demonstrates that the computational methodology we employed had high specificity. We have been studying AR genomic structural variation for several years now, and have tested (and published) use of several SV callers, beginning with Hydra as outlined in our publication in *Oncogene* 2012 Nov 8;31(45):4759-67. We have since moved to use of LUMPY and Delly because these algorithms require both paired-end and split-read support for SV calling. Further, Delly is the SV detection algorithm used in the MSKCC clinical cancer pipeline (MSK-IMPACT: Cheng DT et al., *The Journal of Molecular Diagnostics*, Vol. 17, No. 3, May 2015). We would like to point out that the output from our LUMPY/Delly computational methodology provided many additional calls in which we had less confidence for follow-up (i.e. less than 10 split reads and 10 paired-end reads supporting the SV call, see Supplementary Data 2).

Nevertheless, to directly respond to the reviewers' concern, we implemented Pindel as an alternative SV caller using our AR DNA-seq data as input. Overall, Pindel returned many more calls than our LUMPY/Delly strategy (summarized in the attached spreadsheet for review purposes). Interestingly, there was very little overlap between the output from Pindel and the intersecting LUMPY/Delly calls we had reported in Supplementary Data 2. Closer inspection of these LUMPY/Delly and Pindel overlapping calls revealed that Pindel "missed" the translocation that had been validated by PCR and Sanger sequencing in tumor 05-217F, and which resulted in the novel AR-V discovered in our study (Fig. 5). Pindel also missed several additional AR-GSRs that we had

validated via PCR and Sanger sequencing (summarized in the attached spreadsheet for review purposes). Collectively, we feel these considerations provide considerable support for our focusing on intersecting calls from LUMPY and Delly.

My response 1

Since there is little overlap between Pindel and Delly + Lumpy calls, I am not surprised by the fact that at least one SV discovered by Delly + Lumpy could not be found by Pindel. I suspect the reverse would be the case as well.

More importantly, the authors did not try the other two methods I suggested. Unlike Pindel, or Lumpy or Delly, these tools, e.g. VariationHunter, do take into account repeat regions of the genome. The authors mention Hydra, a tool that reimplements the algorithm introduced by VariationHunter - but this time by the developers of Lumpy. The fact that the authors prefer Lumpy over Hydra does not justify their choice of Lumpy over other available tools - as Hydra's design choices do differ from, e.g., VariationHunter's. Also Delly being used in a clinical pipeline is not a justification for ignoring other tools.

I agree that no available tool is ideal for clonal tumor populations and perhaps a short paragraph can highlight some of the difficulties the authors encountered with available SV detection tools working with short read technologies.

Response to Reviewer #1

Comment

The authors have addressed the major concerns of the study, which is much improved. The findings will be of high impact for the nuclear receptor and prostate cancer fields.

Response

Thank you.

Response to Reviewer #2

Comment

The manuscript by Henzler, Li, Yang et al., focussing on AR structural variants in castration resistant prostate cancer, has been extensively revised in response to the comments from three reviewers. The new analyses are sound and I am happy to recommend publication in Nature Communications.

I agree with the author's approach of focussing on specificity rather than sensitivity of structural variant calls; the variants they report are high-confidence and well validated and are a good starting point of this detailed study.

Response

Thank you.

Response to Reviewer #3

Comment

Since there is little overlap between Pindel and Delly + Lumpy calls, I am not surprised by the fact that at least one SV discovered by Delly + Lumpy could not be found by Pindel. I suspect the reverse would be the case as well.

More importantly, the authors did not try the other two methods I suggested. Unlike Pindel, or Lumpy or Delly, these tools, e.g. VariationHunter, do take into account repeat regions of the genome. The authors mention Hydra, a tool that reimplements the algorithm introduced by VariationHunter - but this time by the developers of Lumpy. The fact that the authors prefer Lumpy over Hydra does not justify their choice of Lumpy over other available tools - as Hydra's design choices do differ from, e.g., VariationHunter's. Also Delly being used in a clinical pipeline is not a justification for ignoring other tools.

I agree that no available tool is ideal for clonal tumor populations and perhaps a short paragraph can highlight some of the difficulties the authors encountered with available SV detection tools working with short read technologies.

Response

The reviewer makes a cogent argument. Based on these considerations, we expanded the second last paragraph of the Discussion section to highlight some of the difficulties we encountered with available SV detection tools while working with short read DNA-seq technology.